# *Sh3bp2* Gain-Of-Function Mutation Ameliorates Lupus Phenotypes in B6.MRL-*Fas^lpr^* Mice

**DOI:** 10.3390/cells8050402

**Published:** 2019-04-30

**Authors:** Akiko Nagasu, Tomoyuki Mukai, Masanori Iseki, Kyoko Kawahara, Shoko Tsuji, Hajime Nagasu, Yasuyoshi Ueki, Katsuhiko Ishihara, Naoki Kashihara, Yoshitaka Morita

**Affiliations:** 1Department of Rheumatology, Kawasaki Medical School, Kurashiki, Okayama 701-0192, Japan; nagasu.a@med.kawasaki-m.ac.jp (A.N.); kyoko.k0925@gmail.com (K.K.); shoko.0513@med.kawasaki-m.ac.jp (S.T.); morita@med.kawasaki-m.ac.jp (Y.M.); 2Department of Immunology and Molecular Genetics, Kawasaki Medical School, Kurashiki, Okayama 701-0192, Japan; miseki@med.kawasaki-m.ac.jp (M.I.); ishihara-im@med.kawasaki-m.ac.jp (K.I.); 3Department of Nephrology and Hypertension, Kawasaki Medical School, Kurashiki, Okayama 701-0192, Japan; HajimeNagasu@kms-ndh.com (H.N.); kashinao@med.kawasaki-m.ac.jp (N.K.); 4Indiana Center for Musculoskeletal Health, Indiana University, Bloomington, IN 46202, USA; uekiy@iu.edu

**Keywords:** SH3 domain–binding protein 2, systemic lupus erythematosus, murine lupus model, Fas, lpr mutation, double-negative T cells, anti-dsDNA antibody, tumor necrosis factor, macrophages, dendritic cells

## Abstract

SH3 domain-binding protein 2 (SH3BP2) is an adaptor protein that is predominantly expressed in immune cells, and it regulates intracellular signaling. We had previously reported that a gain-of-function mutation in SH3BP2 exacerbates inflammation and bone loss in murine arthritis models. Here, we explored the involvement of SH3BP2 in a lupus model. *Sh3bp2* gain-of-function (P416R knock-in; *Sh3bp2^KI/+^*) mice and lupus-prone B6.MRL-*Fas^lpr^* mice were crossed to yield double-mutant (*Sh3bp2^KI/+^Fas^lpr/lpr^*) mice. We monitored survival rates and proteinuria up to 48 weeks of age and assessed renal damage and serum anti-double-stranded DNA antibody levels. Additionally, we analyzed B and T cell subsets in lymphoid tissues by flow cytometry and determined the expression of apoptosis-related molecules in lymph nodes. *Sh3bp2* gain-of-function mutation alleviated the poor survival rate, proteinuria, and glomerulosclerosis and significantly reduced serum anti-dsDNA antibody levels in *Sh3bp2^KI/+^Fas^lpr/lpr^* mice. Additionally, B220^+^CD4^−^CD8^−^ T cell population in lymph nodes was decreased in *Sh3bp2^KI/+^Fas^lpr/lpr^* mice, which is possibly associated with the observed increase in cleaved caspase-3 and tumor necrosis factor levels. *Sh3bp2* gain-of-function mutation ameliorated clinical and immunological phenotypes in lupus-prone mice. Our findings offer better insight into the unique immunopathological roles of SH3BP2 in autoimmune diseases.

## 1. Introduction

Systemic lupus erythematosus (SLE) is an autoimmune disease that affects multiple organs [1] and is characterized by the production of autoantibodies, including an anti-nuclear antibody and an anti-double-stranded DNA (dsDNA) antibody [2,3]. The presence of autoreactive T and B cell clones accounts for the increased production of the autoantibodies [2,3]; therefore, the presence of autoreactive lymphocytes and autoreactive antibodies is a prominent feature of SLE. In addition to the acquired immune system, the innate immune system also contributes to the induction and progression of SLE [4]. The inflammatory cytokines produced by the innate immune cells activate the adaptive immune system and promote tissue damage [4].

The clinical features of SLE are recapitulated in several animal models [5,6,7], one of which involves homozygous Murphy Roths large/lymphoproliferation (lpr) mice (MRL.*Fas^lpr^* mice), which carry a loss-of-function mutation in the death-receptor *Fas/CD95* gene [8]. The loss-of-function mutation in *Fas* results in impaired Fas-mediated apoptosis of autoreactive lymphocytes and subsequent accumulation of these cells [9]. These lupus-prone mice develop massive lymphoproliferation and visceral-organ damage associated with increased production of autoreactive lymphocytes and autoantibodies, such as the anti-dsDNA antibody [7]. The lupus model emphasizes the importance of Fas-mediated peripheral tolerance in SLE pathogenesis [10,11]. In addition to Fas, other pro-apoptotic factors, including tumor necrosis factor (TNF)-related apoptosis-inducing ligand (TRAIL) and TNF, are reportedly involved in the peripheral deletion of pathogenic autoreactive lymphocytes [12,13,14]; however, the detailed mechanisms have not yet been clarified.

SH3 domain-binding protein 2 (SH3BP2) is an adapter protein expressed primarily in immune cells, such as macrophages [15,16], B cells [17,18], and T cells [19]. SH3BP2 regulates immune-cell functions by interacting with various intracellular signaling proteins, including Syk [20,21], phospholipase Cγ [20,22], Vav [23,24], and Src [25,26]. *SH3BP2* mutations are identified as being responsible for the genetic disorder cherubism (OMIM no. 118400), characterized by jaw-bone destruction [27]. We had previously generated *Sh3bp2* cherubism-specific Pro416Arg (P416R) mutation knock-in (KI) mice; the mutation being equivalent to the most common human P418R mutation [15,27]. Analyses of *Sh3bp2* P416R-KI mice revealed enhanced TNF production from activated macrophages [15,16,28,29]. Additionally, *Sh3bp2* gain-of-function mutations reportedly enhance the phagocytic capacity of macrophages [21,30]. Previously, we had reported the involvement of SH3BP2 in the pathogenesis of autoimmune arthritis, with an *Sh3bp2* gain-of-function mutation aggravating joint inflammation and destruction in murine arthritis models [28,31]. However, the pathological roles of SH3BP2 in other immune-mediated diseases have not yet been elucidated.

In this study, we investigated the involvement of SH3BP2 in SLE pathophysiology, using *Sh3bp2* P416R gain-of-function mice and lupus-prone mice carrying the *Fas^lpr^* mutation. Our results demonstrated that *Sh3bp2* gain-of-function mutation improved the survival rate and renal involvement in lupus-prone mice via the reduction in anti-dsDNA antibody titer and autoreactive lymphocytes.

## 2. Materials and Methods

### 2.1. Mice

*Sh3bp2* P416R gain-of-function mutation KI heterozygous (*Sh3bp2^KI/+^*) mice were generated by introducing a Pro-to-Arg mutation into exon 9 of the murine *Sh3bp2* gene, as reported previously [15,31]. B6.MRL-*Fas^lpr^* mice (C57BL/6J background; referred to as *Fas^lpr^* mice) were obtained from the Jackson Laboratory (Bar Harbor, ME, USA). All wild-type (WT) and mutant mice were maintained in the animal facility of Kawasaki Medical School (Okayama, Japan). All mice were housed in groups (2–5 mice/cage) and maintained at 22 °C under a 12 h:12 h light/dark cycle with free access to water and standard laboratory food (MF diet, Oriental Yeast Co., Tokyo, Japan). All animal experiments were approved by the Safety Committee for Recombinant DNA Experiments (Nos. 14-33 and 18-23) and the Institutional Animal Care and Use Committee of Kawasaki Medical School (Nos. 17-042 and 17-131). All experimental procedures were conducted in accordance with the institutional and National Institutes of Health guidelines for the humane use of animals.

### 2.2. Animal Study: Analysis of the Double-Mutant Mice

*Fas^lpr^* mice were crossed with *Sh3bp2^KI/+^* mice (C57BL/6J background) to yield double-mutant mice, including WT (*n* = 8), *Sh3bp2^KI/+^* (*n* = 7), *Fas^lpr/lpr^*(*n* = 12), and *Sh3bp2^KI/+^Fas^lpr/lpr^* (*n* = 8), all of which were monitored until 48 weeks of age. At the end of the observation period, samples of urine, blood, lymph node, spleen, and kidney were collected and used for subsequent analyses.

### 2.3. Western Blot Analysis

Protein expression in the lymph nodes and spleen was determined by western blot, as described previously [28,32]. For preparation of protein samples, tissues were harvested from 48-week-old WT, *Sh3bp2^KI/+^*, *Fas^lpr/lpr^*, and *Sh3bp2^KI/+^Fas^lpr/lpr^* mice immediately after euthanasia and soaked in the RIPA lysis buffer (Sigma-Aldrich, St. Louis, MO, USA) containing a protease inhibitor cocktail (P8340, Sigma-Aldrich), which in turn contains AEBSF, Aprotinin, Bestatin hydrochloride, E-64, Leupeptin hemisulfate salt, and Pepstatin A, and phosphatase inhibitor cocktails (P5726, P0044, Sigma-Aldrich). The tissues were minced using homogenizers. After centrifugation (17,000× *g* for 15 min at 4 °C), supernatants were collected, and protein concentrations were determined using a BCA protein assay kit (Thermo Fisher Scientific, Waltham, MA, USA). Protein samples were resolved by sodium dodecyl sulfate-polyacrylamide gel electrophoresis and transferred to nitrocellulose membranes. After blocking with 5% skim milk in Tris-buffered saline with 0.1% Tween-20, the membranes were incubated with the indicated primary antibodies, followed by incubation with the appropriate horseradish peroxidase (HRP)-conjugated species-specific secondary antibodies. Bands were detected using SuperSignal West Dura or Femto chemiluminescent substrate (Thermo Fisher Scientific) and visualized using an ImageQuant LAS-4000 (GE Healthcare, Little Chalfont, UK). Actin was used as a loading control to normalize the amount of protein. The antibodies used in this study were as follows: anti-SH3BP2 (clone 1E9; Abnova, Taipei City, Taiwan), anti-caspase 3 (D3R6Y; Cell Signaling Technology, Danvers, MA, USA), and anti-Actin (A2066; Sigma-Aldrich).

### 2.4. Urine Protein Assessment

Spot urine samples were collected every 8 weeks from the age of 8 weeks. Protein levels in urine were individually evaluated using urine test strips (Albustix; Siemens Healthineers, Tokyo, Japan). Levels of proteinuria were semiquantitatively graded from 0 to 3 (0: < 30 mg/dL; 1+: 30–99 mg/dL; 2+: 100–299 mg/dL; 3+: > 300 mg/dL).

### 2.5. Histopathologic Assessment

Tissue samples were fixed in 4% paraformaldehyde for 2 days and then embedded in paraffin. Kidney sections (2 µm) were stained with periodic acid-Schiff. Glomerulosclerosis was assessed in a blinded manner, semiquantitatively, using the glomerulosclerosis index (GSI) as follows [33,34]: a score of 0 was assigned to normal glomeruli, 1 for up to 25% involvement, 2 for up to 50% involvement, 3 for up to 75% involvement, and 4 for > 75% sclerosis. The GSI was calculated using the following formula: GSI = [(1 × N1) + (2 × N2) + (3 × N3) + (4 × N4)]/(N0 + N1 + N2 + N3 + N4)(1)
where NX is the number of glomeruli, with each given score, for a given section. At least 15 glomeruli were randomly selected from each mouse, and the mean score was calculated therefrom.

### 2.6. Serum Anti-dsDNA Antibody Measurement

Anti-dsDNA antibody levels in serum samples were measured using an enzyme-linked immunosorbent assay (ELISA) kit (Shibayagi, Gunma, Japan). Diluted sera (1:100) were incubated on dsDNA-coated ELISA plates for 2 h at 25 °C; after washing, the plates were incubated with HRP-conjugated goat anti-mouse IgG for 2 h at 25 °C. Tetramethylbenzidine was used for detection, and optical density at 450 nm (OD_450_) was measured using a microplate reader (Varioskan Flash; Thermo Fisher Scientific). Concentrations of anti-dsDNA antibodies (IgG) were calculated and expressed as mU/mL.

### 2.7. Serum Immunoglobulin Measurement

Concentrations of isotype-specific immunoglobulins in serum were measured with an ELISA kit SBA Clonotyping System-C57BL/6-AP (Southern Biotech, Birmingham, AL, USA). Each well in the 96-well plate was incubated with goat anti-mouse immunoglobulin (10 μg/mL) as a capture reagent at 4 °C overnight. Wells were blocked with 1% bovine serum albumin (BSA) in phosphate-buffered saline (PBS) at 4 °C overnight. Diluted serum samples were added to the capture antibody-coated wells and incubated for 1 h at 25 °C with gentle shaking. The wells were then incubated with alkaline-phosphatase-labeled detection antibodies for 1 h at 25 °C. After adding p-nitrophenyl phosphate substrate, optical densities were measured at 405 nm by a microplate reader (Varioskan Flash), and the relative quantity of isotype-specific immunoglobulins was determined.

### 2.8. Flow Cytometry

The subsets of immune cells in the bone marrow, spleen, and lymph nodes were analyzed with a flow cytometer (FACSCanto II; BD Biosciences, Franklin, NJ, USA). To block FcγR, single-cell suspensions were incubated with the anti-CD16/CD32 antibody (2.4G2; BD Biosciences) on ice for 10 min before staining with the indicated monoclonal antibodies. The following monoclonal antibodies were used in this study: anti-IgM (RMM-1), anti-IgD (11-26c.2a), anti-CD19 (6D5), anti-mouse CD45R/B220 (RA3-6B2), anti-CD43 (S11), anti-CD21/CD35 (7E9), anti-CD23 (B3B4), anti-CD4 (RM4-4), anti-CD8a (53-6.7), and anti-CD25 (PC61, all from BioLegend, San Diego, CA, USA), and anti-CD3ε (eBioscience, San Diego, CA, USA); all antibodies were conjugated with fluorochrome. Dead cells were excluded by 7-aminoactinomycin D (7-AAD; BioLegend) staining. In most samples, a minimum of 3 × 10^4^ events was evaluated, with all data analyzed using FlowJo software (version 9.9.5; BD Biosciences).

### 2.9. In Vivo Immunization and Measurement of Serum Immunoglobulins Levels

To evaluate antibody production against thymus-independent or -dependent antigens, serum immunoglobulin levels were quantified after immunization of the mice, as previously described [17,18,35]. For thymus-independent antigen experiments, 8-week-old WT and *Sh3bp2^KI^^/+^* mice were immunized intraperitoneally with 50 μg of trinitrophenol conjugated to Ficoll (TNP-Ficoll; Biosearch Technologies, Petaluma, CA, USA) in sterile PBS, and blood samples were collected at 0-, 7-, and 14-days post-immunization. For thymus-dependent antigen experiments, 8-week-old WT and *Sh3bp2^KI^^/+^* mice were immunized intraperitoneally with 100 μg of TNP-keyhole limpet hemocyanin (KLH; Biosearch Technologies) in sterile PBS, which was pre-incubated in Imject Alum adjuvant (Thermo Fisher Scientific). Mice were given booster injections with the same dose at day 14, and blood samples were collected at 0-, 7-, 14-, and 21-days post-immunization. For anti-TNP ELISA, each well of the 96-well plate was coated with 100 μL of 10 μg/mL TNP-BSA (Biosearch Technologies) in PBS at 4 °C overnight. Plates were then blocked with 1% BSA in PBS at 4 °C overnight, and the relative quantity of TNP-specific antibodies in each serum sample was determined using an isotype-specific ELISA kit (SBA Clonotyping System-C57BL/6-AP; Southern Biotech).

### 2.10. Real-Time Quantitative PCR (qPCR) Analysis

The qPCR analysis was performed as described previously [28,32]. Total RNA samples were extracted from lymph nodes and cultured cells using RNAiso Plus (Takara Bio, Shiga, Japan) and solubilized in RNase-free water. The cDNA was synthesized using a Prime Script RT reagent kit (Takara Bio). The qPCR reactions were performed using SYBR^®^ Green PCR master mix (Takara Bio) with the StepOnePlus system (Thermo Fisher Scientific). Gene-expression levels relative to hypoxanthine phosphoribosyltransferase were calculated by the ΔΔCt method and normalized to baseline controls, as indicated in each experiment. All qPCR reactions yielded products with single-peak dissociation curves. Primers used in this study are listed in Table 1.

### 2.11. Bone-Marrow-Derived Dendritic Cell (BMDC) Culture

A crude population of DCs was generated in vitro from mouse bone marrow cells, as described previously [36], with some modifications. Briefly, bone marrow cells were flushed from the tibia and femur of the mice; the collected cells were plated at a density of 1 × 10^6^/mL and cultured for 9 days in 5% CO_2_ at 37 °C in RPMI 1640 medium containing 10% heat-inactivated fetal bovine serum (FBS), 10 ng/mL recombinant mouse granulocyte-macrophage colony stimulating factor (GM-CSF; PeproTech, Rocky Hill, NJ, USA), and 5 ng/mL mouse interleukin (IL)-4 (PeproTech). The yielded cells were used as crude BMDCs (CD11c^+^ cells; 70–80%) for subsequent experiments.

### 2.12. TNF Measurement by ELISA

TNF concentrations in culture supernatants were measured by sandwich ELISA (R&D Systems, Minneapolis, MN, USA) [28]. Each well of 96-well plate was coated with the goat anti-mouse TNF antibody as a capture reagent and incubated at 4 °C overnight; thereafter, the wells were washed and blocked with 1% BSA in PBS at 4 °C overnight. Samples and standards were added to the wells and incubated for 2 h at 25 °C. After washing, the wells were incubated with a biotinylated goat anti-mouse TNF antibody as a detection reagent for 2 h at 25 °C and then incubated with streptavidin-HRP substrate solution. OD_450_ was measured using a microplate reader (Varioskan Flash), and TNF concentration in each sample was calculated based on a standard curve.

### 2.13. Bone Marrow-Derived Macrophage (BMM) Culture

Isolation and culture of primary bone marrow cells were performed, as previously described [28,31,37]. Briefly, bone marrow cells were isolated from the long bones of 10-week-old female mice and cultured on Petri dishes for 2 h at 37 °C under 5% CO_2_. Non-adherent bone marrow cells were re-seeded on culture plates at a density of 1 × 10^6^ cells/mL and then incubated for 2 days in α-minimum essential medium (α-MEM) containing 10% heat-inactivated FBS and 25 ng/mL recombinant mouse macrophage colony stimulating factor (M-CSF; PeproTech). After the 2-day pre-culture, the yielded BMMs were stimulated with lipopolysaccharide (LPS) in the presence of M-CSF. RNA samples were isolated from BMMs at the indicated time points, and subjected to gene-expression analysis.

### 2.14. Phagocytosis Assay

#### 2.14.1. Preparation of Apoptotic Cells

Jurkat cells were cultured in RPMI 1640 containing 10% FBS, 50 μM 2-mercaptoethanol, and 2 mM L-glutamine. Cells were suspended in PBS and irradiated under an ultraviolet lamp for 20 min (> 80% cells were trypan blue-positive). Apoptotic Jurkat cells were resuspended in RPMI 1640 medium and incubated overnight, followed by labeling with IncuCyte pHrodo fluorescent dye, according to manufacturer’s instructions (Essen BioScience, Ann Arbor, MI, USA). The pHrodo dye being pH sensitive allows actual detection of ingested apoptotic cells based on the increased fluorescence emission in the acidic environment of phagocyte-derived phagosomes [38].

#### 2.14.2. Measurement of Phagocytic Activity

Non-adherent bone marrow cells were isolated as described above (Section 2.13) and cultured in α-MEM containing 10% heat-inactivated FBS and 25 ng/mL M-CSF. On day 9, the yielded BMMs were collected and seeded onto 96-well plates (2 × 10^4^ cells/well). On day 10, pHrodo-labeled apoptotic Jurkat cells (5 × 10^4^ cells) were added to each well, and BMMs and apoptotic cells were co-incubated at 37 °C under 5% CO_2_. Fluorescence intensity (excitation/emission wavelength: 560/585 nm) was measured at 4 h, 24 h, and 48 h of co-culture using a microplate reader (Varioskan Flash). As a negative control, pHrodo-labeled Jurkat cells were incubated without BMMs for the same period, and values of the wells with labeled Jurkat cells alone were subtracted as background.

### 2.15. Statistical Analysis

All data represent the mean ± standard deviation. Statistical analyses were performed using the two-tailed unpaired Student’s *t*-test to compare two groups and one-way analysis of variance (Tukey’s post hoc test) to compare three or more groups. Survival rates and incidences of proteinuria were analyzed by Fisher’s exact test. GraphPad Prism v7 (GraphPad Software Inc., San Diego, CA, USA) was used for all statistical analyses, and a *P* < 0.05 was considered statistically significant.

## 3. Results

### 3.1. Sh3bp2 Gain-Of-Function Mutation Improves the Survival Rate of Lupus-Prone Mice

To explore the involvement of SH3BP2 in SLE pathogenesis, we generated a double-mutant mouse carrying an *Sh3bp2* P416R-KI mutation along with the *Fas^lpr^* mutation. The *Sh3bp2* P416R mutation contributes to gain-of-function by increasing SH3BP2 stability [39,40]. In *Sh3bp2* P416R-KI mice, SH3BP2 protein accumulates in the cytoplasm, with SH3BP2-mediated pathways subsequently becoming highly activated [28,31]. We used these mutant mice as an *Sh3bp2* gain-of-function model. 

We first analyzed SH3BP2 protein expression and found it elevated in the lymph nodes and spleens of the *Sh3bp2^KI/+^* and *Sh3bp2^KI/+^Fas^lpr/lpr^* mice, compared to those in WT and *Fas^lpr/lpr^* mice (Figure 1a). This indicated that *Sh3bp2^KI/+^* mutation increased SH3BP2 protein expression, regardless of the *Fas^lpr^* mutation, and that *Fas^lpr^* mutation did not affect SH3BP2 levels.

We observed WT, *Sh3bp2^KI/+^*, *Fas^lpr/lpr^,* and *Sh3bp2^KI/+^Fas^lpr/lpr^* mice until the age of 48 weeks, and found that body weights were comparable between *Fas^lpr/lpr^* and *Sh3bp2^KI/+^Fas^lpr/lpr^* mice (Figure 1b), and that the extent of splenomegaly in *Sh3bp2^KI/+^Fas^lpr/lpr^* mice tended to be milder than that in *Fas^lpr/lpr^* mice (Figure 1c). The severity of lymphadenopathy was comparable between *Fas^lpr/lpr^* and *Sh3bp2^KI/+^Fas^lpr/lpr^* mice (data not shown). Interestingly, *Sh3bp2^KI/+^Fas^lpr/lpr^* mice exhibited a significantly improved survival rate as compared to that of *Fas^lpr/lpr^* mice (41.7% in *Fas^lpr/lpr^* vs. 75.0% in *Sh3bp2^KI/+^Fas^lpr/lpr^* mice at 48 weeks) (Figure 1d). This data together indicated that the *Sh3bp2* gain-of-function mutation ameliorated pathological conditions in the lupus-prone mice. The improved phenotypes in the lupus model were opposite to the aggravated phenotypes caused by the *Sh3bp2* gain-of-function mutation in murine arthritis models, which we had previously reported [28,31].

### 3.2. Sh3bp2 Gain-Of-Function Mutation Improves Renal Involvement in Lupus-Prone Mice

To determine how the *Sh3bp2* gain-of-function mutation improves the survival rate of the *Fas^lpr^* lupus-prone mice, we examined renal involvement, which is a characteristic feature of lupus-prone mice [7]. Most *Fas^lpr/lpr^* mice exhibited profound proteinuria (>100 mg/dL; indicated as ≥2+), whereas the incidence of severe proteinuria was significantly lower in *Sh3bp2^KI/+^Fas^lpr/lpr^* mice relative to that in *Fas^lpr/lpr^* mice (100% in *Fas^lpr/lpr^* vs. 50.0% in *Sh3bp2^KI/+^Fas^lpr/lpr^* mice at 48 weeks) (Figure 2a). Histologic analysis of the kidney revealed the development of glomerulosclerotic lesions in *Fas^lpr/lpr^* mice, with the lesions being milder in *Sh3bp2^KI/+^Fas^lpr/lpr^* mice (Figure 2b and Appendix A), and represented as a decreased glomerulosclerosis score (Figure 2c). These findings suggested that the *Sh3bp2* gain-of-function mutation attenuated the progression of renal damage, resulting in the improved survival rate in *Sh3bp2^KI/+^Fas^lpr/lpr^* mice.

### 3.3. Sh3bp2 Gain-Of-Function Mutation Reduces the Production of Serum Anti-dsDNA Antibody

Autoantibody production is a critical process associated with the development of organ damage in patients with SLE and lupus-prone mice [2,3,6]. To examine how the *Sh3bp2* gain-of-function mutation modulates the immunological phenotypes in *Fas^lpr^* lupus-prone mice, we determined serum concentrations of the anti-dsDNA antibody (IgG). *Fas^lpr/lpr^* mice exhibited elevated levels of serum anti-dsDNA, whereas the *Sh3bp2* gain-of-function mutation significantly suppressed serum anti-dsDNA levels in lupus-prone mice at 32 and 48 weeks (Figure 3a), indicating that the *Sh3bp2* gain-of-function mutation negatively regulated autoantibody production in concert with the improved survival rate (Figure 1d). Additionally, we examined the serum levels of immunoglobulin subclasses (IgM, IgG1, IgG2b, IgG2c, and IgG3). The finding that all of these were elevated in *Fas^lpr/lpr^* mice reflected the aberrant activation of antibody-producing cells. Moreover, we found that the *Sh3bp2* gain-of-function mutation reduced the excessive production of immunoglobulins (Figure 3b).

Next, we determined in vivo antibody production against thymus-independent and -dependent antigens. WT and *Sh3bp2^KI/+^* mice were immunized with a thymus-independent antigen (TNP-Ficoll) and a thymus-dependent antigen (TNP-KLH), followed by measurement of serum antibody levels. Antibody levels were comparable between WT and *Sh3bp2^KI/+^* mice (Figure 4), suggesting that the *Sh3bp2* gain-of-function mutation did not significantly alter the antibody-producing capacities of B cells.

To further analyze the mechanisms associated with reduced in vivo immunoglobulin production in lupus-prone mice, we determined the differentiation and maturation status of B cells in the bone marrow and spleen by flow cytometry. In both bone marrow cells and splenic cells, we observed no significant difference in the patterns of B cell subsets between *Fas^lpr/lpr^ and Sh3bp2^KI/+^Fas^lpr/lpr^* mice (Figure 5a–c), hence suggesting that B cell differentiation and maturation processes are not significantly altered by *Sh3bp2* mutation.

### 3.4. Aberrant Accumulation of the B220^+^CD4^−^CD8^−^ T cell Population is Ameliorated in Sh3bp2^KI/+^Fas^lpr/lpr^ Mice

Accumulation of B220^+^CD4^−^CD8^−^ T cells, referred to as double-negative T (DNT) cells, is a prominent feature of *Fas^lpr^* lupus-prone mice [7,41,42]. It is thought to be caused by impaired peripheral tolerance [11,43]. 

To determine the effect of SH3BP2 on T cells, we examined T cell subsets in lymph nodes from lupus-prone mice and found the ratios of CD4^+^ T cells, CD8^+^ T cells, and CD4^+^CD25^+^ T cells to be comparable between *Fas^lpr/lpr^* and *Sh3bp2^KI/+^Fas^lpr/lpr^* mice (Figure 6a). However, the number of DNT cells in the lymph nodes of *Fas^lpr/lpr^* mice was significantly elevated relative to that in WT mice. Interestingly, the number of DNT cells in the lymph nodes was significantly decreased in the *Sh3bp2^KI/+^Fas^lpr/lpr^* mice, compared to that in *Fas^lpr/lpr^* mice (Figure 6a,b). These findings suggested that the aberrant accumulation of autoreactive lymphocytes in lupus-prone mice was partly relieved by the *Sh3bp2* gain-of-function mutation.

### 3.5. Sh3bp2^KI/+^Fas^lpr/lpr^ Mice Exhibit Increased Activation of Apoptosis-Inducing Cascades in the Lymph Nodes

We hypothesized that DNT cells might have been effectively deleted in *Sh3bp2^KI/+^Fas^lpr/lpr^* mice, although deletion of the DNT cells was largely disrupted due to the *Fas* mutation. To examine whether an apoptotic pathway is activated in the lymph nodes of *Sh3bp2^KI/+^Fas^lpr/lpr^* mice, we determined the levels of caspase-3. We found elevated levels of cleaved caspase-3, an active form of caspase-3, in the lymph nodes of *Sh3bp2^KI/+^Fas^lpr/lpr^* mice (Figure 7a), which suggested increased activation of apoptotic processes in *Sh3bp2^KI/+^Fas^lpr/lpr^* mice. Additionally, we examined the expression of other apoptosis-related genes in the lymph nodes and found reduced *Fas* mRNA expression in *Fas^lpr/lpr^* mice, consistent with a previous report [44]. Moreover, we found that mRNA levels of both *Fas* and *Fas ligand* (*FasL*) were comparable between *Fas^lpr/lpr^* and *Sh3bp2^KI/+^Fas^lpr/lpr^* mice (Figure 7b), whereas *Tnf* and *Tnfr1* mRNA levels in the lymph nodes tended to be elevated in *Sh3bp2^KI/+^Fas^lpr/lpr^* mice. However, there was no significant difference in mRNA expression of *Tnfr2*, *Trail*, *Trailr2*, and *Dr3* between the *Fas^lpr/lpr^* and *Sh3bp2^KI/+^Fas^lpr/lpr^* mice (Figure 7b).

Next, we examined TNF expression in DC and macrophage cultures. We found TNF mRNA and protein levels to be elevated in DCs from mice with *Sh3bp2^KI/+^* mutation (Figure 8a,b), and *Tnf* mRNA to be elevated in LPS-stimulated macrophages from *Sh3bp2^KI/+^* and *Sh3bp2^KI/+^Fas^lpr/lpr^* mice (Figure 8c).

### 3.6. Phagocytic Activity Was Unaltered in Sh3bp2^KI/+^Fas^lpr/lpr^ Mice

Disrupted clearance of self-antigens is reportedly attributed to the development of autoantibodies in SLE [45,46]. Since the *Sh3bp2* gain-of-function mutation reportedly enhanced the phagocytic capacity of macrophages [21,30], we examined macrophage phagocytosis activity. BMMs were isolated from the indicated mice, and phagocytic capacities of apoptotic cells were determined. Fluorescence-labeled apoptotic Jurkat cells were co-cultured with BMMs, and fluorescence intensities derived from engulfed cells were determined. No detectable difference was found in the phagocytic activity of BMMs from *Fas^lpr/lpr^* and *Sh3bp2^KI/+^Fas^lpr/lpr^* mice (Figure 8d).

## 4. Discussion

In this study, we examined the involvement of the adaptor protein SH3BP2 in SLE pathophysiology using *Sh3bp2* gain-of-function mice. Our results revealed that excess SH3BP2 protein ameliorated immunological and pathological phenotypes in *Fas^lpr^* lupus-prone mice and that the *Sh3bp2* gain-of-function mutation prolonged the survival and reduced the renal involvement in these mice. The improved phenotypes were associated with suppressed production of the anti-dsDNA antibody and decreased accumulation of DNT cells in the lymph nodes. Notably, we observed elevated levels of cleaved caspase-3, which plays a central role in apoptosis progression [47] in the lymph nodes of *Sh3bp2^KI/+^Fas^lpr/lpr^* mice.

Accumulation of autoreactive lymphocytes is a crucial step in SLE initiation [1]. The Fas-FasL system is a regulatory mechanism involved in attenuating pathogenic autoreactive lymphocytes [10,11]. *Fas^lpr^* mice, which possess a loss-of-function mutation in *Fas*, clearly demonstrate the importance of Fas-mediated peripheral immune tolerance in the pathogenesis of lupus [2,3]. In the present study, our findings of decreased DNT cells and increased cleaved caspase-3 levels in lymph nodes of *Sh3bp2^KI/+^Fas^lpr/lpr^* mice led us to speculate that the *Sh3bp2* gain-of-function mutation at least partially rescued the impaired deletion of autoreactive lymphocytes in a Fas-independent manner. In addition to Fas and FasL, other pro-apoptotic factors also contribute to attenuating pathogenic autoreactive lymphocytes. TRAIL and TNF induce Fas-independent apoptosis [12,13,14], and we observed no difference in *Trail* mRNA levels in lymph nodes between *Fas^lpr/lpr^* and *Sh3bp2^KI/+^Fas^lpr/lpr^* mice in the present study, whereas *Tnf* mRNA expression tended to increase in the lymph nodes of *Sh3bp2^KI/+^Fas^lpr/lpr^* mice. Additionally, we revealed increased production of TNF by myeloid cells from *Sh3bp2^KI/+^Fas^lpr/lpr^* mice. These findings suggested that increased TNF expression in *Sh3bp2^KI/+^Fas^lpr/lpr^* myeloid cells might have contributed to the deletion of DNT cells, resulting in improved lupus-like phenotypes.

TNF negatively regulates lupus induction. In clinical settings, administration of TNF inhibitors can trigger the development of autoantibodies and even lupus manifestation (i.e., drug-induced lupus) in patients with rheumatoid arthritis and inflammatory bowel diseases [48,49]. Although the mechanisms associated with TNF-inhibitor-induced lupus remain uncertain, several mechanisms have been hypothesized. According to one, TNF inhibitors induce apoptosis in inflammatory cells, followed by the excessive release of endogenous immunostimulatory molecules, including self-DNA [50]. As per another, TNF inhibitors promote autoimmunity by elevating susceptibility to infections or deviating cytokine production from T helper (Th)2 to Th1 cells [51,52]. In addition to clinical evidence, in vivo murine models demonstrated the protective effect of TNF in lupus development, showing improvement in survival rate and renal damage after the administration of TNF to lupus-prone NZB/NZW F1 mice [53]. TNFR1 deficiency reportedly accelerates lymphadenopathy and anti-dsDNA antibody production in *Fas^lpr^* lupus-prone mice [14]. Moreover, TNF deficiency aggravates immune-complex deposition in the kidney associated with increased number of plasmacytoid DCs and elevated levels of type I interferons in a pristane-induced lupus model [54]. These findings in human SLE and murine lupus models support our hypothesis that increased TNF expression in *Sh3bp2* gain-of-function mutant cells might ameliorate clinical and immunological phenotypes observed in lupus-prone mice. Contrary to the protective effect of TNF in the development of SLE, some reports show that TNF contributes to the progression of SLE. Administration of anti-TNF antibody is reported to attenuate lupus phenotypes in a murine lupus model with *Fasl* mutation [55] and in an experimental lupus model [56]. Since TNF has dualistic suppressive and promotive effects on lupus condition [57], timing and duration of TNF exposure or inhibition seem to be critical.

SH3BP2 plays multiple roles in various immune cells [58]; however, based on our previous findings [15,28], *Sh3bp2* gain-of-function mutation appears to dominantly regulate the functions of myeloid cells, rather than of lymphocytes. We had previously shown that the *Sh3bp2* gain-of-function mutation aggravates joint inflammation and destruction in a human TNF-transgenic model and a collagen-induced arthritis model [28,31]. In the latter model, *Sh3bp2^KI/+^* mutation activated macrophages and osteoclasts, whereas antibody production and T cell activation were not detectably altered. These findings together suggested that the *Sh3bp2^KI/+^* mutation dominantly affects the functions of myeloid cells, but not lymphocytes, in this animal model. The myeloid-cell-dominant effects of the *Sh3bp2* gain-of-function mutation agree with the phenotypes observed in homozygous *Sh3bp2^KI/KI^* mice, which spontaneously develop systemic organ inflammation [15]. The spontaneous inflammation in *Sh3bp2^KI/KI^* mice is dominantly mediated by macrophages, since increased CD11b-positive cells infiltrate into the inflamed lesions [15]. The inflammatory phenotypes of *Sh3bp2^KI/KI^* mice are not rescued by T and B cell defects induced by Rag-1 deficiency [15]. These findings together support our hypothesis that macrophage activation and their increased TNF production in *Sh3bp2* gain-of-function mutant cells may ameliorate clinical and immunological phenotypes of the lupus-prone mice.

Interestingly, the cell types affected by SH2BP2 mutation vary depending on gain- or loss-of-function. As described, the gain-of-function mutation dominantly affects myeloid cells, whereas loss-of-function mutation affects both lymphocytes and myeloid cells. We had previously shown that SH3BP2 deficiency dramatically suppresses the production of anti-type II collagen antibody without affecting T cell response to type II collagen in a collagen-induced arthritis model [37]. Additionally, B cell-dominant effects of SH3BP2 deficiency were reported by two groups, revealing that SH3BP2 deficiency impairs the optimal responses of B cells, but not of T cells [17,18]. SH3BP2 deficiency appears to impair B cell differentiation, maturation, and proliferation through disrupted response to the B cell receptor [17]. In addition to its effect on B cells, SH3BP2 deficiency affects the functions of macrophages by impairing cytokine production, phagocytic activity [21,30], as well as osteoclast formation [25,37]. In a TNF-transgenic arthritis model, SH3BP2 deficiency decreased joint destruction via direct suppression of osteoclastogenesis [37]. These findings suggested that SH3BP2 loss-of-function has a broader impact on the cellular functions of various immune cells, whereas *Sh3bp2* gain-of-function dominantly affects the function of myeloid cells. This might be due to the different regulatory mechanisms associated with SH3BP2 expression in each immune cell. Further analyses using cell-specific mutant mice would be required to clarify these points.

Effects of the *Sh3bp2* gain-of-function mutation on disease processes in the lupus model differed from those observed in previous arthritis models [28,31]. The *Sh3bp2* gain-of-function mutation rescued lupus phenotypes while exacerbating joint inflammation and destruction in murine arthritis models [28,31]. The differential effects of *Sh3bp2* gain-of-function mutation between arthritis and lupus models might be attributed to the distinct pathological impact of TNF under arthritis and lupus conditions. TNF is a cytokine critical for the development and progression of arthritis, as shown by progressive arthritis in murine TNF-transgenic mice [59], and dramatic therapeutic effect of TNF inhibitors on human rheumatoid arthritis [60]. On the other hand, the contribution of TNF in SLE progression is limited, as shown by the minor effect of TNF inhibitors on patients with SLE [61]. Conversely, inhibition of TNF raises the risk of inducing lupus-like manifestations [49], as described here. Therefore, increased TNF expression in *Sh3bp2* gain-of-function mutant cells might aggravate inflammation in arthritic conditions while alleviating immunological and pathological symptoms in lupus-like conditions.

Impaired phagocytic activity of macrophages reportedly contributes to SLE pathogenesis [62,63]. Suppression of phagocytosis decreases the clearance of self-DNA from apoptotic cells and subsequently increases levels of self-DNA capable of triggering the production of anti-DNA antibodies [62,63]. SH3BP2 reportedly affects the phagocytic activity of macrophages [21,30]. The phagocytic activity of U937 cells, a human macrophage cell line, is enhanced following transduction of the human *SH3BP2* P418R mutant, which is equivalent to the murine *Sh3bp2* P416R mutation [21]. These findings prompted our initial hypothesis that elevated phagocytic activity in macrophages harboring the *Sh3bp2* gain-of-function mutation might augment the clearance of self-antigens and subsequently suppress aberrant autoimmune responses. To test this hypothesis, we examined the phagocytosis of apoptotic cells in BMM culture; however, we did not observe an increased phagocytic activity of *Sh3bp2^KI/+^Fas^lpr/lpr^* macrophages (Figure 8d). Despite this result, we cannot completely exclude the possibility that elevated phagocytic activity of *Sh3bp2^KI/+^Fas^lpr/lpr^* macrophages modifies the clearance of self-DNA leading to subsequent anti-dsDNA production. The phagocytic activity might vary depending on experimental settings, as described in previous reports of elevated phagocytic activity of *Sh3bp2^KI/+^* macrophages in experiments using fluorescence-labeled zymosan and beads [21,30]. Further analyses using various experimental settings would be required to determine whether SH3BP2-mediated dysregulation of phagocytic activity affects lupus induction and progression.

Our findings highlighted the pathological roles of macrophages in SLE pathophysiology. Macrophages regulate the autoimmune system by modulating multiple pathways, not just through TNF expression and phagocytic activity. Abnormal expression of monocyte-surface markers, such as FcγRIII, CD40, CD244, CR4, Siglec-1, and Siglec-4, have been reported in monocytes from patients with SLE [62,64,65,66]; therefore, dysregulation of monocyte surface molecules could skew optimal immune responses, including pathogenic cytokine and chemokine expression [62].

In our study, however, some questions still remain to be answered. First, why splenomegaly and lymphadenopathy were not significantly retrieved in *Sh3bp2^KI/+^Fas^lpr/lpr^* mice is unclear. Detailed analysis at different time points might reveal improved splenomegaly and lymphadenopathy in *Sh3bp2^KI/+^Fas^lpr/lpr^* mice. Alternatively, *Sh3bp2* gain-of-function mutation might promote the growth of stromal cells in the tissues, considering a recent report that suggested SH3BP2 regulates the growth of stromal tumor [67]. Second, whether *Sh3bp2* gain-of-function mutation could affect the proliferative and functional characteristics of monocytes and macrophages need to be revealed. Although we had previously reported that macrophages from *Sh3bp2^KI/+^* mice are hyper-activated in response to M-CSF and various TLR ligands [15,16], the proliferative and functional characteristics had not been determined in vivo or in vitro. Determination of these characteristics would facilitate further understanding of the improved phenotypes of *Sh3bp2^KI/+^Fas^lpr/lpr^* mice. Third, whether DNT cells would be deleted by the TNF expressed by myeloid cells is not understood clearly. Although we proposed the mechanisms, direct evidence of apoptosis in DNT cells is lacking. To ascertain our proposed mechanisms, further analyses would be required, e.g., determination of apoptosis in DNT cells of lymph nodes or in vitro analysis to determine the apoptotic pathway in DNT cells.

Our current study indicated that excess SH3BP2 protein suppressed immunological and clinical lupus phenotypes. These findings led us to consider the modulation of SH3BP2 expression as a potential therapeutic approach for SLE, since elevated SH3BP2 protein levels might improve immunological abnormality and organ involvement in SLE. From this standpoint, tankyrase would likely be a suitable candidate for drug targeting based on its role as a poly(ADP-ribose) polymerase involved in regulating SH3BP2 stability [39,40]. Tankyrase ribosylates SH3BP2 and subsequently induces SH3BP2 ubiquitination and degradation [39,40]. We had previously reported that treatment with tankyrase inhibitors induces SH3BP2 accumulation in macrophages [32]. Therefore, tankyrase inhibition might potentially ameliorate immunological abnormalities associated with lupus induction and progression. Since the administration of tankyrase inhibitors induces bone loss [32], the development of a drug-delivery system would be necessary to advance this therapeutic approach.

## 5. Conclusions

In conclusion, we demonstrated that the *Sh3bp2* gain-of-function mutation ameliorated clinical and immunological phenotypes in lupus-prone mice and that these improved phenotypes were associated with decreased autoantibody production and autoreactive lymphocytes, which were likely induced by activated TNF-producing myeloid cells. Our findings revealed unique immunopathological consequences associated with the *Sh3bp2* gain-of-function mutation in a lupus model and highlighted the essential roles of myeloid cells in SLE pathogenesis.

## Figures and Tables

**Figure 1 cells-08-00402-f001:**
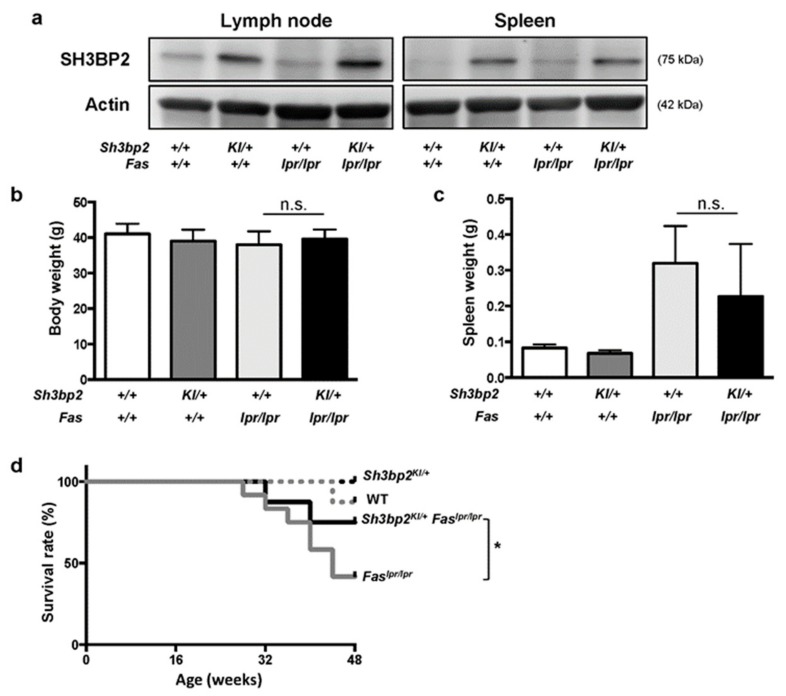
*Sh3bp2* gain-of-function mutation improves the survival rate of *Fas^lpr^* lupus-prone mice. (**a**) Immunoblot analysis for SH3BP2. Protein samples were collected from lymph nodes and spleens of WT, *Sh3bp2^KI/+^*, *Fas^lpr/lpr^*, and *Sh3bp2^KI/+^Fas^lpr/lpr^* mice. Actin was used as a loading control. (**b**–**d**) WT (*n* = 8), *Sh3bp2^KI/+^* (*n* = 7), *Fas^lpr/lpr^* (*n* = 12), and *Sh3bp2^KI/+^Fas^lpr/lpr^* mice (*n* = 8) were monitored until the age of 48 weeks. At the end of the observation period, body weight (**b**) and spleen weight (**c**) were measured. (**d**) Survival rates of the mice. Values are presented as the mean ± SD. Note: * *p* < 0.05; n.s. = not significant. SH3BP2, SH3 domain-binding protein 2; WT, wild-type; KI, knock-in.

**Figure 2 cells-08-00402-f002:**
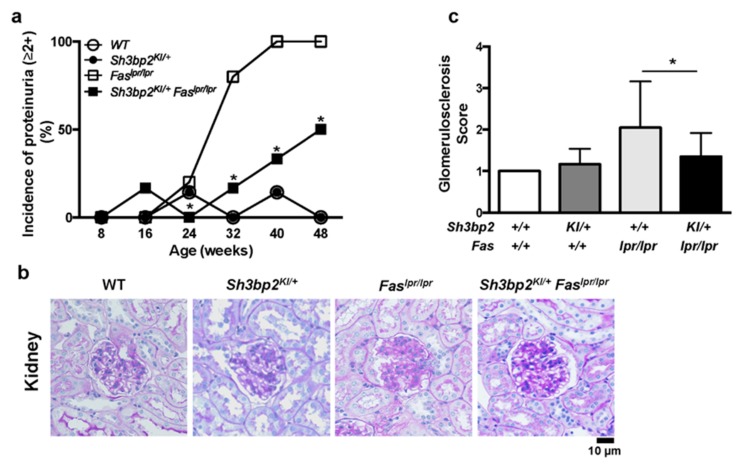
*Sh3bp2* gain-of-function mutation improves renal involvement in *Fas^lpr^* lupus-prone mice. (**a**) Incidence of proteinuria. WT (*n* = 7), *Sh3bp2^KI/+^* (*n* = 7), *Fas^lpr/lpr^* (*n* = 5), and *Sh3bp2^KI/+^Fas^lpr/lpr^* mice (*n* = 6) were monitored until the age of 48 weeks, and levels of proteinuria were monitored every 8 weeks and graded as follows: 0, <30 mg/dL; 1+, 30–99 mg/dL; 2+, 100–299 mg/dL; 3+, >300 mg/dL. Incidence of proteinuria (≥2+) is presented. (**b**) Images of periodic acid-Schiff (PAS)-stained kidney sections from 48-week-old mice. Original magnification, 400×. Bar, 10 μm. (**c**) Renal glomerulosclerosis score. Severity of glomerulosclerosis was graded from 0 to 4, with 15 to 20 glomeruli graded per mouse. Values are presented as the mean ± SD. Note: * *p* < 0.05. SH3BP2, SH3 domain-binding protein 2; WT, wild-type; KI, knock-in

**Figure 3 cells-08-00402-f003:**
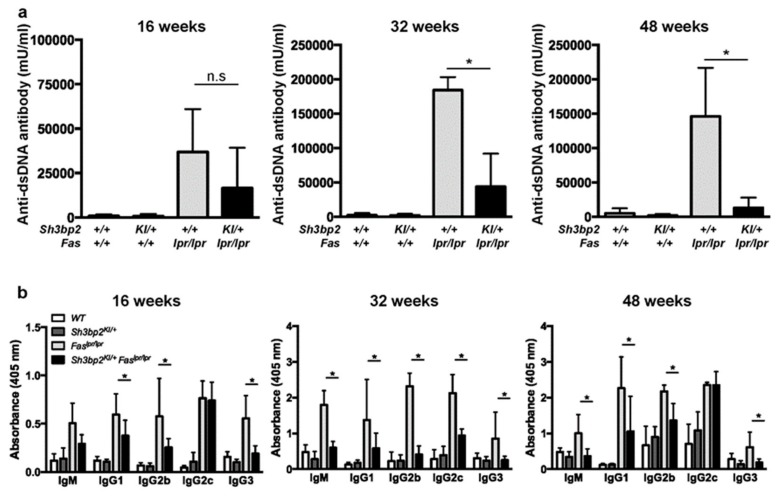
*Sh3bp2* gain-of-function mutation suppresses aberrant anti-dsDNA antibody and immunoglobulin production in *Fas^lpr^* lupus-prone mice. (**a**,**b**) Serum samples were collected from WT (*n* = 7), *Sh3bp2^KI/+^* (*n* = 7), *Fas^lpr/lpr^* (*n* = 5), and *Sh3bp2^KI/+^Fas^lpr/lpr^* mice (*n* = 5) at 16, 32, and 48 weeks of age. Levels of anti-dsDNA antibody (**a**) and each immunoglobulin subclass (**b**) were determined by ELISA. Values are presented as the mean ± SD. * *p* < 0.05; n.s. = not significant. Note: dsDNA, double-stranded DNA; SH3BP2, SH3 domain-binding protein 2; WT, wild-type; KI, knock-in; ELISA, enzyme-linked immunosorbent assay.

**Figure 4 cells-08-00402-f004:**
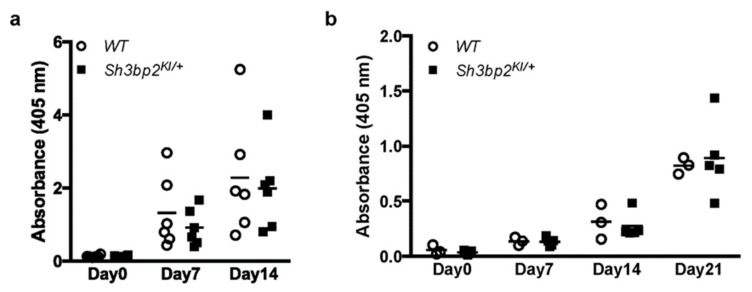
Antibody production against thymus-independent (TI) and thymus-dependent (TD) antigens. Antibody production was comparable between WT (*n* = 6) and *Sh3bp2^KI/+^* (*n* = 6) mice. (**a**) For TI antigen experiments, 8-week-old WT and *Sh3bp2^KI/+^* mice were immunized with TNP-Ficoll. Blood samples were collected at 0-, 7-, and 14-days post-immunization, and levels of the anti-TNP antibody (IgG3) in serum (1:100 dilution) were measured by ELISA. (**b**) For TD antigen experiments, 8-week-old WT (*n* = 3) and *Sh3bp2^KI/+^* (*n* = 5) mice were immunized with TNP-KLH and given a booster injection with the same dose at day 14. Blood samples were collected at 0-, 7-, 14-, and 21-days post-immunization, and levels of the anti-TNP antibody (IgG1) in serum (1:1000 dilution) were measured by ELISA. Note: SH3BP2, SH3 domain-binding protein 2; WT, wild-type; KI, knock-in; TNP, trinitrophenol; KLH, keyhole limpet hemocyanin; ELISA, enzyme-linked immunosorbent assay

**Figure 5 cells-08-00402-f005:**
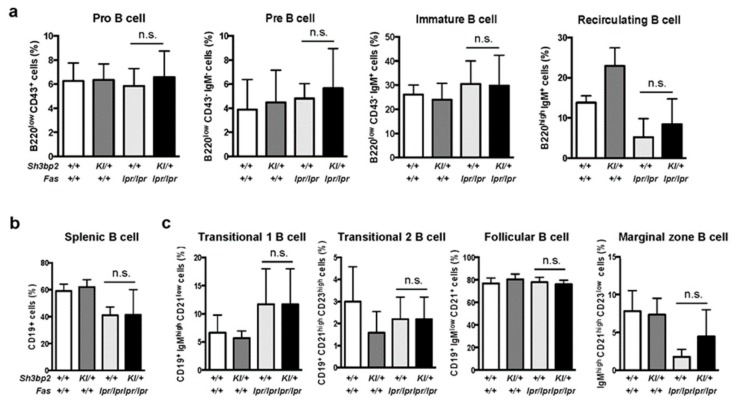
B cell differentiation and maturation are unaltered in *Sh3bp2^KI/+^Fas^lpr/lpr^* mice. (**a–c**) B cell subsets were analyzed in the bone marrow (**a**) and spleen (**b**,**c**) by flow cytometry. Cells were collected from WT (*n* = 7), *Sh3bp2^KI/+^* (*n* = 7), *Fas^lpr/lpr^* (*n* = 5), and *Sh3bp2^KI/+^Fas^lpr/lpr^* mice (*n* = 6) at 48 weeks of age, and the suspended cells were stained with fluorochrome-labeled antibodies against IgM, IgD, CD43, and B220 for bone marrow cells (**a**) and IgM, IgD, CD19, CD23, and CD21/35 for splenic cells (**b**,**c**). All cells were initially gated as 7-AAD-negative single cells, followed by being gated as lymphocytes (**a**,**b**) or CD19^+^ cells (**c**). Values are presented as the mean ± SD. Note: * *p* < 0.05; n.s. = not significant. SH3BP2, SH3 domain-binding protein 2; WT, wild-type; KI, knock-in; 7-AAD, 7-aminoactinomycin D.

**Figure 6 cells-08-00402-f006:**
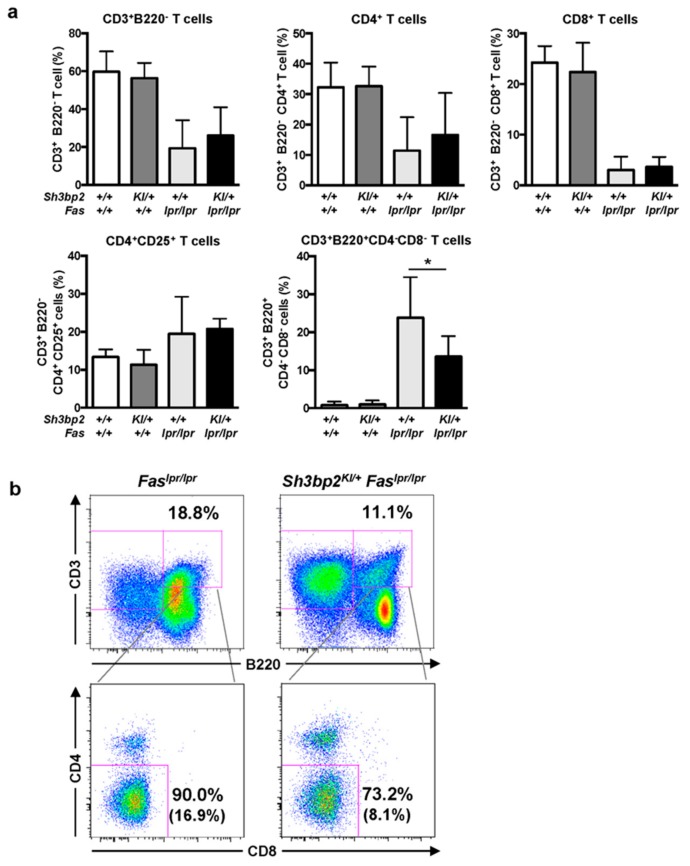
CD3^+^B220^+^CD4^−^CD8^−^ DNT cells are reduced in the lymph nodes of *Sh3bp2^KI/+^Fas^lpr/lpr^* mice. (**a**,**b**) Lymph node cells were collected from WT (*n* = 7), *Sh3bp2^KI/+^* (*n* = 7), *Fas^lpr/lpr^* (*n* = 5), and *Sh3bp2^KI/+^Fas^lpr/lpr^* mice (*n* = 6) at 48 weeks of age, and T cell subsets were stained with fluorochrome-labeled antibodies against CD3, CD4, CD8, CD25, and B220. (**a**) The ratio of T cell subsets was analyzed by flow cytometry. All cells were gated as 7-AAD-negative single cells, followed by being gated as lymphocytes. Values are presented as the mean ± SD; * *P* < 0.05; n.s. = not significant. (**b**) Representative flow cytometry plots of DNT cells in the lymph nodes. Flow cytometry shows a decreased proportion of DNT cells in the lymphocyte fraction of *Sh3bp2^KI/+^Fas^lpr/lpr^* cells. The number in parentheses indicates the percentage of DNT cells in total lymphocytes. Note: DNT, double-negative T cell; SH3BP2, SH3 domain-binding protein 2; WT, wild-type; KI, knock-in; 7-AAD, 7-aminoactinomycin D.

**Figure 7 cells-08-00402-f007:**
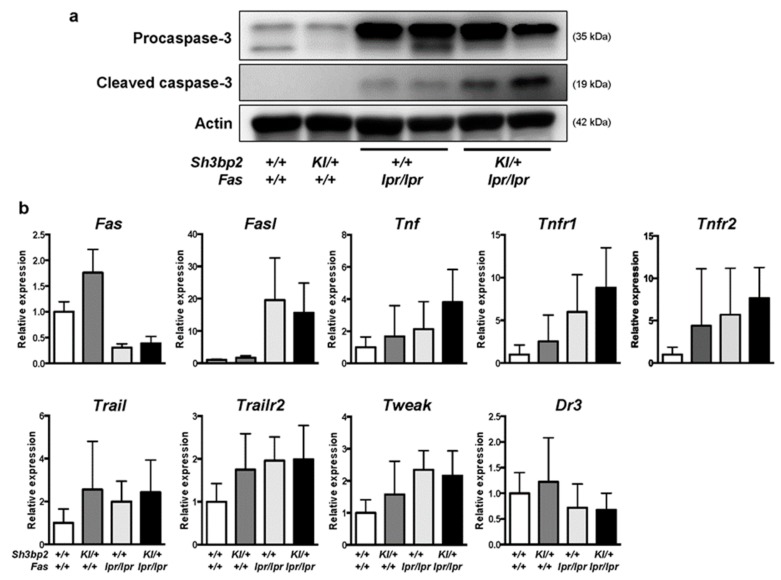
Levels of cleaved caspase-3 are elevated in the lymph nodes of *Sh3bp2^KI/+^Fas^lpr/lpr^* mice. (**a**) Images of immunoblot for procaspase-3 and cleaved caspase-3. Protein samples were collected from the lymph nodes of WT (*n* = 4), *Sh3bp2^KI/+^* (*n* = 4), *Fas^lpr/lpr^* (*n* = 4), and *Sh3bp2^KI/+^Fas^lpr/lpr^* (*n* = 4) mice. Representative images for procaspase-3 and cleaved caspase-3 are presented. Actin was used as a loading control. (**b**) qPCR analysis of apoptosis-related genes. RNA samples were collected from the lymph nodes of WT (*n* = 7), *Sh3bp2^KI/+^* (*n* = 7), *Fas^lpr/lpr^* (*n* = 5), and *Sh3bp2^KI/+^Fas^lpr/lpr^* (*n* = 6) mice at 48 weeks of age, and gene-expression levels relative to that of *Hprt* were determined and normalized against levels in WT samples. Values are presented as the mean ± SD. Note: SH3BP2, SH3 domain-binding protein 2; WT, wild-type; KI, knock-in; Fasl, Fas ligand; Tnf, tumor necrosis factor; Tnfr1, TNF receptor 1; Tnfr2, TNF receptor type 2; Trail, TNF-related apoptosis-inducing ligand; Trailr2, TRAIL receptor 2; Tweak, TNF-like weak inducer of apoptosis; Dr3, death receptor 3.

**Figure 8 cells-08-00402-f008:**
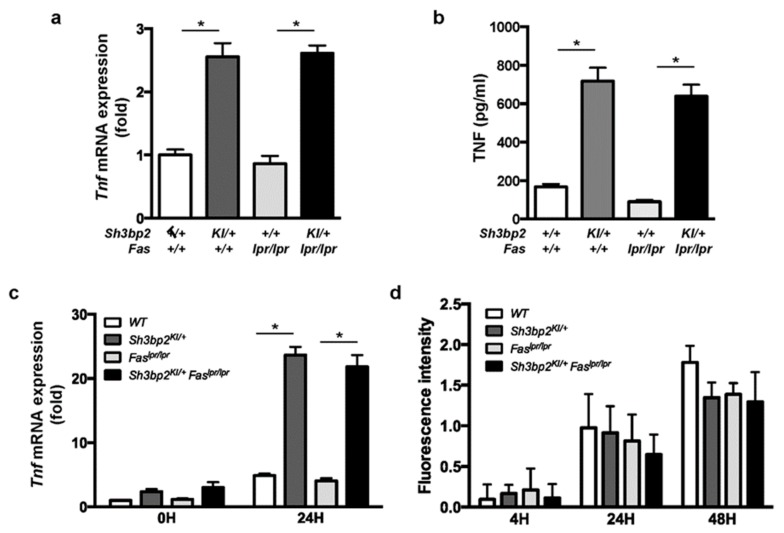
TNF is highly expressed in the *Sh3bp2* gain-of-function mutant DCs and macrophages. (**a**,**b**) Bone marrow cells were isolated from 14- to 15-week-old WT, *Sh3bp2^KI/+^*, *Fas^lpr/lpr^*, and *Sh3bp2^KI/+^Fas^lpr/lpr^* mice and pre-cultured with GM-CSF (20 ng/mL) and IL-4 (5 ng/mL) for 8 days; resulting BMDCs were used for the experiments. (**a**) *Tnf* mRNA levels relative to that of *Hprt* were determined by qPCR. (**b**) TNF protein levels in culture supernatants. Culture supernatants were collected at the end of BMDC culture, and TNF levels were determined by ELISA. (**c**) *Tnf* mRNA expression in BMMs. Bone marrow cells were isolated from 10- to 12-week-old WT, *Sh3bp2^KI/+^*, *Fas^lpr/lpr^*, and *Sh3bp2^KI/+^Fas^lpr/lpr^* mice and stimulated with M-CSF (25 ng/mL) for 2 days, after which the yielded BMMs were treated with LPS (1 ng/mL) in the presence of M-CSF (25 ng/mL). *Tnf* mRNA levels relative to that of *Hprt* were determined by qPCR. (**d**) Phagocytic capacity of BMMs. Apoptotic Jurkat cells were labeled with pH-sensitive fluorescent dye and co-cultured with BMMs, followed by the measurement of fluorescence intensity derived from the engulfed apoptotic cells. Values are presented as the mean ± SD. Note: * *P* < 0.05. SH3BP2, SH3 domain-binding protein 2; WT, wild-type; KI, knock-in; BMDC, bone marrow-derived dendritic cell; BMM, bone marrow-derived macrophage; GM-CSF, granulocyte-macrophage colony stimulating factor; M-CSF, macrophage colony stimulating factor; IL-4, interleukin-4; LPS, lipopolysaccharide; Hprt, hypoxanthine phosphoribosyltransferase.

**Table 1 cells-08-00402-t001:** The quantitative PCR (qPCR) primers used in this study.

Target Genes	Gene Accession Number	Sequence (5′-3′)		Amplicon Size (bp)
*Fas*	NM_007987.2	Forward: Reverse:	taaaccagacttctactgcgattct gggttccatgttcacacga	73
*Fasl*	NM_010177.4	Forward: Reverse:	aaaaagagccgaggagtgtg attccagagggatggacctt	66
*Tnf*	NM_013693.2	Forward: Reverse:	catcttctcaaaattcgagtgaca tgggagtagacaaggtacaaccc	175
*Tnfr1*	NM_011609.4	Forward: Reverse:	ggaaagtatgtccattctaagaacaa agtcactcaccaagtaggttcctt	76
*Tnfr2*	NM_011610.3	Forward: Reverse:	gaggcccaagggtttcag ggcttccgtgggaagaat	64
*Trail*	NM_ 009425.2	Forward: Reverse:	ggctgcaagtctgcattggg cctgcagctgcttcatctcgt	192
*Trailr2*	NM_ 020275.4	Forward: Reverse:	ccacaacacggaacctggca tgtctgtgaggcccaactgc	167
*Tweak*	NM_ 011614.3	Forward: Reverse:	caggatggagcacaagcag ggctggagctgttgattttg	73
*Dr3*	NM_ 001291010.1	Forward: Reverse:	ccctggcttatcccagact agatgccagaggagttccaa	97
*Hprt*	NM_013556.2	Forward: Reverse:	tcctcctcagaccgctttt cctggttcatcatcgctaatc	90

*Fasl*, Fas ligand; *Tnf*, tumor necrosis factor; *Tnfr1*, TNF receptor type 1; *Tnfr2*, TNF receptor type 2; *Trail*, TNF-related apoptosis-inducing ligand; *Trailr2*, TRAIL receptor 2; *Tweak*, TNF-like weak inducer of apoptosis; *Dr3*, death receptor 3; *Hprt*, hypoxanthine phosphoribosyltransferase.

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
