# Peer review of "Sh3bp2* Gain-Of-Function Mutation Ameliorates Lupus Phenotypes in B6.MRL-*Fas^lpr^* Mice"

_cells, 2019, doi:10.3390/cells8050402_

Round 1

Reviewer 1 Report

The authors describe a mutation in SH3BP2 that partially rescues the phenotype of the lupus prone B6.MRL-Faslpr mouse.  The presentation of the data and writing of the manuscript are exemplary and the claims are generally supported.  Although the numbers of mice used are not exhaustive, the depth and breadth of assays used give a solid backing to their main hypothesis that the increased production of TNF caused by their mutation is generally beneficial in the model of lupus. 

1 "Unexpected" effect of SH3BP2

The authors claim (ln 26) that the mutation unexpectedly improved lupus however TNF production (which has previously been shown for this mutation) is associated with reduced disease in lupus.  While it could be argued that there is not a strong consensus on the mechanism(s) of this effect, the potential for an amelioration of disease in lupus is not wholly unexpected.

2 Proteinuria

Figure 2A plots the data for severe proteinuria, which was defined as 100-299mg/dL.  The authors may wish to expand this figure element to include data on the percentage of mice with milder proteinuria.

3 Figure 3B

It is understandable that the authors wish to have the same axes for the 3 graphs in Figure 3B however the bars in the 16 weeks graph are very low.  The authors may wish to alter the axis on the 16 weeks graph to improve readability.

4 Figure 7

Although Tnfr1 is thought to be the main Tnfr through which apoptosis is initiated, the authors should measure Tnfr2 or at least mention why they did not measure it in the discussion.

Minor points

ln18 "dominantly".  Should this be "predominantly"?

ln 95 State the protease inhibitors used.

ln 100 State the percentage of Tween20.

ln 218-220 This is very similar to ln 200-203.  Please consider consolidating.

ln 589 "notion".  Should this be "hypothesis"?

Author Response

We thank the reviewers for their comments on our manuscript, cells-479458, and for the invitation to submit the revised manuscript. We have carefully considered the comments offered by the three reviewers. In revising the manuscript, we have made substantial changes in response to all issues raised in the initial review. A marked copy of the revised manuscript is uploaded as a “Cells-479458 SLE-SH3BP2 Revision Marked copy” with the changes addressing the reviewers’ comments shown in red for ease of identification by the reviewers. The revisions made in response to the comments of each reviewer are as follows:

Reviewer#1

The authors describe a mutation in SH3BP2 that partially rescues the phenotype of the lupus-prone B6.MRL-Faslpr mouse.  The presentation of the data and writing of the manuscript are exemplary and the claims are generally supported.  Although the numbers of mice used are not exhaustive, the depth and breadth of assays used give a solid backing to their main hypothesis that the increased production of TNF caused by their mutation is generally beneficial in the model of lupus. 

1 "Unexpected" effect of SH3BP2

The authors claim (ln 26) that the mutation unexpectedly improved lupus however TNF production (which has previously been shown for this mutation) is associated with reduced disease in lupus.  While it could be argued that there is not a strong consensus on the mechanism(s) of this effect, the potential for an amelioration of disease in lupus is not wholly unexpected.

Response: We thank the reviewer for pointing this out. Actually, we intended to say “The improved phenotypes in the lupus model were opposite to the aggravated phenotypes caused by the Sh3bp2 gain-of-function mutation in murine arthritis models, which we had previously reported [28,31].”, as described at the end of Results, sub-section 3.1. As suggested by the reviewer, we agree that the word “unexpectedly” may be misleading, and hence, have deleted the word “unexpectedly” from the Abstract and Results sections.

2 Proteinuria

Figure 2A plots the data for severe proteinuria, which was defined as 100-299mg/dL.  The authors may wish to expand this figure element to include data on the percentage of mice with milder proteinuria.

Response: We thank the reviewer for the suggestion. Even among wild-type mice, most mice exhibited mild proteinuria (1+) due to relatively high tubular proteinuria compared to humans. We had originally defined “severe proteinuria” as the condition when the mice exhibited 2+ proteinuria. To avoid confusion, we have now deleted the word “severe proteinuria” and replaced it consistently with “proteinuria (2+)”.

3 Figure 3B

It is understandable that the authors wish to have the same axes for the 3 graphs in Figure 3B however the bars in the 16 weeks graph are very low.  The authors may wish to alter the axis on the 16 weeks graph to improve readability.

Response: We agree with the reviewer in this issue, and have replaced the 2 graphs at 16 weeks of age by new ones with modified Y axes. We hope this revision would improve its readability.

4 Figure 7

Although Tnfr1 is thought to be the main Tnfr through which apoptosis is initiated, the authors should measure Tnfr2 or at least mention why they did not measure it in the discussion.

Response: As suggested by the reviewer, we have performed an additional experiment to measure Tnfr2 mRNA expression. The data have now been added to Figure 7b.

Minor points

ln18 "dominantly".  Should this be "predominantly"?

Response: We are sorry for this mistake; we have corrected it now.

ln 95 State the protease inhibitors used.

Response: We appreciate the reviewer for this suggestion. We have now added the requisite information in Methods, sub-section 2.3; we used a protease inhibitor cocktail (P8340, Sigma-Aldrich), which contains AEBSF, Aprotinin, Bestatin hydrochloride, E-64, Leupeptin hemisulfate salt, and Pepstatin A. We have also added product numbers of the protease inhibitor and phosphatase inhibitor cocktails for reference.

ln 100 State the percentage of Tween20.

Response: The percentage of Tween20 in the TBS-T buffer was 0.1%. We have now added the information in the revised manuscript.

ln 218-220 This is very similar to ln 200-203.  Please consider consolidating.

Response: We appreciate the reviewer for pointing at the redundant description of BMM culture. We have now consolidated the description for BMM culture.

ln 589 "notion".  Should this be "hypothesis"?

Response: As suggested by the reviewer, we have now changed the word “notion” to “hypothesis.”

Reviewer 2 Report

The authors shown that a gain of function in the adaptor protein Sh3bp2 decreased some of the SLE symptoms in the SLE mouse model. Specifically, a Sh3bp2 gain of function improved survival rate in the SLE mouse model and the incidence of renal glomerulosclerosis. Additionally, Sh3bp2 gain of function mice shown a decrease in the levels of anti-dsDNA antibodies but also of the total levels of IgGs and IgM. In a cellular level, only differences were found in the amount of double negative T cells but not in of the B cell population analyzed.

Although the consequences of a Sh3bp2 gain of function are clear, some questions should be addressed by the authors.

1-Figure 2-At least 3 images from 3 different mice per group should be added. In addition, in the text more information about the score of glomerulosclerosis has to be added.

2-Figure 5 and 6 (differences in cell populations):

2a: The authors analyzed the differentiation and maturation of B cell in spleen and bone marrow but they do not analyzed the levels of activated B cells, plasmablast or plasma cells in the mice. The differences in the levels of antibodies may be related to changes in the levels of these B cell populations.

2b: In the discussion (paragraph 639-644) the author’s emphasized the importance of monocyte and macrophages in the pathology of SLE but, in this study, the levels and characteristics of these populations had not been analyzed. Data about these cells should be added.

3- Figure 7: The western blot of procaspase-3 and cleaved caspase 3 have to be presented complete. In addition, in the figure legend, it had to be indicated how many mice had been analyzed.

4- Figure 8d: There was not differences between the BMMs phagocytic capacities of apoptotic Jurkat, but it will be interesting to analyze if there are differences in the clearance of autoantibodies by the BMMs of the 4 different genotypes.

Author Response

We thank the reviewers for their comments on our manuscript, cells-479458, and for the invitation to submit the revised manuscript. We have carefully considered the comments offered by the three reviewers. In revising the manuscript, we have made substantial changes in response to all issues raised in the initial review. A marked copy of the revised manuscript is uploaded as a “Cells-479458 SLE-SH3BP2 Revision Marked copy” with the changes addressing the reviewers’ comments shown in red for ease of identification by the reviewers. The revisions made in response to the comments of each reviewer are as follows:

Reviewer#2

The authors shown that a gain of function in the adaptor protein Sh3bp2 decreased some of the SLE symptoms in the SLE mouse model. Specifically, a Sh3bp2 gain of function improved survival rate in the SLE mouse model and the incidence of renal glomerulosclerosis. Additionally, Sh3bp2 gain of function mice shown a decrease in the levels of anti-dsDNA antibodies but also of the total levels of IgGs and IgM. In a cellular level, only differences were found in the amount of double negative T cells but not in of the B cell population analyzed.

Although the consequences of a Sh3bp2 gain of function are clear, some questions should be addressed by the authors.

1-Figure 2-At least 3 images from 3 different mice per group should be added. In addition, in the text more information about the score of glomerulosclerosis has to be added.

Response: We would like to thank the reviewer for the constructive suggestions to improve our manuscript. As suggested by the reviewer, we have now prepared additional images of the glomeruli, and provided them in a supplemental file (Figure A1). We have also added a description in the Results section.

2-Figure 5 and 6 (differences in cell populations):

2a: The authors analyzed the differentiation and maturation of B cell in spleen and bone marrow but they do not analyzed the levels of activated B cells, plasmablast or plasma cells in the mice. The differences in the levels of antibodies may be related to changes in the levels of these B cell populations.

Response: We appreciate the reviewer for this suggestion. Although we have not examined the levels of activated B cells and plasma cells, there is a possibility that they might have decreased in Sh3bp2KI/+Faslpr/lpr mice. We are currently engaged in exploring this possibility. However, such studies will obviously require substantial additional time to complete. Therefore, we plan to finalize it in the next report.

2b: In the discussion (paragraph 639-644) the author’s emphasized the importance of monocyte and macrophages in the pathology of SLE but, in this study, the levels and characteristics of these populations had not been analyzed. Data about these cells should be added.

Response: We agree that our in vivo experiments did not investigate the population of monocytes and macrophages, which would be important to support our hypothesis. We would like to explore this in our next study. In the Discussion section of the revised manuscript, we have described this unsolved issue and stated the necessity of further analyses to determine the proliferative and functional characteristics of monocytes and macrophages.

“Second, whether Sh3bp2 gain-of-function mutation could affect the proliferative and functional characteristics of monocytes and macrophages need to be revealed. Although we had previously reported that macrophages from Sh3bp2KI/+ mice are hyper-activated in response to M-CSF and various TLR ligands [15,16], the proliferative and functional characteristics had not been determined in vivo or in vitro. Determination of these characteristics would facilitate further understanding of the improved phenotypes of Sh3bp2KI/+Faslpr/lpr mice.”

3- Figure 7: The western blot of procaspase-3 and cleaved caspase 3 have to be presented complete. In addition, in the figure legend, it had to be indicated how many mice had been analyzed.

Response: We thank the reviewer for pointing this out. We have herein attached the original images depicting both procaspase-3 and cleaved caspase-3 in a single image. For the western blot, while we took a short-exposure image to determine the levels of procaspase-3, we preferred a long-exposure image to determine the levels of cleaved caspase-3, since cleaved caspase-3 was a small fraction of total caspase-3. In that duration, the bands of procaspase-3 had already saturated, and the levels of procaspase-3 were not suitable to compare across genotypes in the image. Therefore, we took its image at a short-exposure time to compare the levels of procaspase-3 across genotypes. We would prefer to retain the images as they are, so that the readers can easily compare the levels of procaspase-3 and cleaved caspase-3 across genotypes. If the editors still opt for procaspase-3 and cleaved caspase-3 in a single image, we can replace the current images with the long-exposure image.

              To evaluate protein expression, we collected 4 protein samples from each genotype; we have added the information in the figure legend.

4- Figure 8d: There was not differences between the BMMs phagocytic capacities of apoptotic Jurkat, but it will be interesting to analyze if there are differences in the clearance of autoantibodies by the BMMs of the 4 different genotypes.

Response: We agree with the reviewer in this regard; however, we have not examined the phagocytic capacities of autoantibodies now. In future experiments, we plan to determine the phagocytic capacities of macrophages against autoantibodies as well as autoantigens.

Reviewer 3 Report

Nagasu et al. investigated the role of SH3BP2 in the pathogenesis of systemic lupus erythematosus (SLE) using a mouse lupus model (double-mutant (Sh3bp2KI/+Faslpr/lpr) mice) by evaluating the survival rate, proteinuria, renal damage, serum anti-double-stranded DNA antibody levels, B and T cell subsets in lymphoid tissues, and the expression of apoptosis-related molecules in lymph nodes of these mice. Overall, this study is interesting, and the data presented in this study are informative. However, several concerns are raising, as listed below. I hope the comments will be helpful to improve the quality of this study.

1. SLE is an autoimmune as well as inflammatory disease. Please, briefly introduce the inflammation and the relevance of inflammatory responses with autoimmunity in the very first paragraph of Introduction chapter.

2. Please provide the protocols number approved by the Institutional Animal Care and Use Committee (IACUC) for the animal study.

3. The authors do not need to provide information about annealing temperature for qPCR in Table 1. Please remove it.

4. Result 1, Fig. 1c: It is generally known that splenomegaly is observed in the inflammatory/autoimmune diseases, such as rheumatoid arthritis. As expected, the spleen weight of Sh3bp2+/+Faslpr/lpr mice was markedly increased, however, the spleen weight of Sh3bp2KI/+Faslpr/lpr mice was not significantly different from that of Sh3bp2+/+Faslpr/lpr mice. Please discuss the possibility of this observation in the discussion chapter.

5. Result 2, Fig. 2a: It seems that the incidence of severe proteinuria of WT and Sh3bp2KI/+ is exactly the same. I was just wondering if the data between the two groups are really the same or misplotted.

6. Result 5: The authors hypothesized that DNT cells might have been effectively deleted in Sh3bp2KI/+Faslpr/lpr mice, and examined the apoptosis pathway in the lymph nodes of these mice. However, to properly examine the authors’ hypothesis, the authors should examine the apoptosis pathway in the DNT cells (CD3+B220+CD4-CD8- T cells) after isolating these cells from the Sh3bp2KI/+Faslpr/lpr mice.

7. Result 5, Fig. 7b: The authors stated that whereas Tnf and Tnfr1 mRNA levels in the lymph nodes tended to be elevated in Sh3bp2KI/+Faslpr/lpr mice (line 445 - 446). The tendency of mRNA expression of both Tnf and Tnfr1 seem to be a little bit increased in the Sh3bp2KI/+Faslpr/lpr mice, however, it does not seem to be statistically different between Sh3bp2+/+Faslpr/lpr and Sh3bp2KI/+Faslpr/lpr mouse groups, indicating that Tnf and Tnfr1 mRNA levels between two groups are not significantly different and that the authors’ hypothesis that apoptosis process in Sh3bp2KI/+Faslpr/lpr mice is increased is not correct based on these results.

8. Result 5, Fig. 8b: TNF is one of the most popular pro-inflammatory cytokines expressed and secreted from the macrophages under inflammatory condition. Based on the authors’ hypothesis, Sh3bp2KI/+Faslpr/lpr ameliorates autoimmunity and is anti-inflammatory, however, TNF production was increased in these mice compared to the Sh3bp2+/+Faslpr/lpr mice. Although the authors provided reference 48, the authors still need to explain the possibility of this observation which is contradictory to the fact well established in this field.

9. Result 5, Fig. 8d: The reviewer strongly recommends the authors to examine the phagocytosis activity of BMMs in various time points (shorter than 24 h or longer than 48 h) since it has been reported in many studies that active phagocytosis of macrophages occurs within 1 h.

10. The labeling of some figures is broken (Fig. 3a – c). Please make a correction properly.

11. There are some typos and grammatical errors. Please go over the entire manuscript carefully and correct all typo and grammatical errors.

Author Response

We thank the reviewers for their comments on our manuscript, cells-479458, and for the invitation to submit the revised manuscript. We have carefully considered the comments offered by the three reviewers. In revising the manuscript, we have made substantial changes in response to all issues raised in the initial review. A marked copy of the revised manuscript is uploaded as a “Cells-479458 SLE-SH3BP2 Revision Marked copy” with the changes addressing the reviewers’ comments shown in red for ease of identification by the reviewers. The revisions made in response to the comments of each reviewer are as follows:

Reviewer#3

Nagasu et al. investigated the role of SH3BP2 in the pathogenesis of systemic lupus erythematosus (SLE) using a mouse lupus model (double-mutant (Sh3bp2KI/+Faslpr/lpr) mice) by evaluating the survival rate, proteinuria, renal damage, serum anti-double-stranded DNA antibody levels, B and T cell subsets in lymphoid tissues, and the expression of apoptosis-related molecules in lymph nodes of these mice. Overall, this study is interesting, and the data presented in this study are informative. However, several concerns are raising, as listed below. I hope the comments will be helpful to improve the quality of this study.

1. SLE is an autoimmune as well as inflammatory disease. Please, briefly introduce the inflammation and the relevance of inflammatory responses with autoimmunity in the very first paragraph of Introduction chapter.

Response: We appreciate the reviewer for such positive comments and suggestions to improve the manuscript further. Following the reviewer’s suggestion, we have now added the relevant description in the Introduction section and have cited a review paper as a new reference (Ref. #4).

4. Moulton, V.R.; Suarez-Fueyo, A.; Meidan, E.; Li, H.; Mizui, M.; Tsokos, G.C. Pathogenesis of Human Systemic Lupus Erythematosus: A Cellular Perspective. Trends in molecular medicine 2017, 23, 615-635, doi:10.1016/j.molmed.2017.05.006.

2. Please provide the protocols number approved by the Institutional Animal Care and Use Committee (IACUC) for the animal study.

Response: The numbers in parenthesis (17-042 and 17-131) in the initial manuscript are the protocol numbers approved by the Institutional Animal Care and Use Committee. In the initial manuscript, we had used “Animal Research Committee” instead of “the Institutional Animal Care and Use Committee”. However, we have now replaced with the accurate information now.

3. The authors do not need to provide information about annealing temperature for qPCR in Table 1. Please remove it.

Response: As suggested by the reviewer, we have now removed the information about annealing temperatures from Table 1.

4. Result 1, Fig. 1c: It is generally known that splenomegaly is observed in the inflammatory/autoimmune diseases, such as rheumatoid arthritis. As expected, the spleen weight of Sh3bp2+/+Faslpr/lpr mice was markedly increased, however, the spleen weight of Sh3bp2KI/+Faslpr/lpr mice was not significantly different from that of Sh3bp2+/+Faslpr/lpr mice. Please discuss the possibility of this observation in the discussion chapter.

Response: As mentioned by the reviewer, the spleen weight of Sh3bp2KI/+Faslpr/lpr mouse was not significantly different from that of Sh3bp2+/+Faslpr/lpr mouse. To explain this observation, we have now added the following description in the Discussion section.

In our study, however, some questions still remain to be answered. First, why splenomegaly and lymphadenopathy were not significantly retrieved in Sh3bp2KI/+Faslpr/lpr mice is unclear. Detailed analysis at different time points might reveal improved splenomegaly and lymphadenopathy in Sh3bp2KI/+Faslpr/lpr mice. Alternatively, Sh3bp2 gain-of-function mutation might promote the growth of stromal cells in the tissues, considering a recent report that suggested SH3BP2 regulates the growth of stromal tumor [67].”

67. Serrano-Candelas, E.; Ainsua-Enrich, E.; Navines-Ferrer, A.; Rodrigues, P.; Garcia-Valverde, A.; Bazzocco, S.; Macaya, I.; Arribas, J.; Serrano, C.; Sayos, J., et al. Silencing of adaptor protein SH3BP2 reduces KIT/PDGFRA receptors expression and impairs gastrointestinal stromal tumors growth. Molecular oncology 2018, 12, 1383-1397, doi:10.1002/1878-0261.12332.

5. Result 2, Fig. 2a: It seems that the incidence of severe proteinuria of WT and Sh3bp2KI/+ is exactly the same. I was just wondering if the data between the two groups are really the same or misplotted.

Response: We are very grateful to the reviewer for noting this. We had written wrong numbers of mice in the figure legend. For the figure of proteinuria (Fig. 2a), we had included the mice that we observed till the age of 48 weeks. The correct numbers are WT (n = 7), Sh3bp2KI/+ (n = 7), Faslpr/lpr (n = 5), and Sh3bp2KI/+Faslpr/lpr mice (n = 6). We have changed the numbers in the figure legend now.

6. Result 5: The authors hypothesized that DNT cells might have been effectively deleted in Sh3bp2KI/+Faslpr/lpr mice, and examined the apoptosis pathway in the lymph nodes of these mice. However, to properly examine the authors’ hypothesis, the authors should examine the apoptosis pathway in the DNT cells (CD3+B220+CD4-CD8- T cells) after isolating these cells from the Sh3bp2KI/+Faslpr/lpr mice.

Response: We agree with the reviewer that it would be important to validate our proposed mechanisms. We are now planning to determine the levels of apoptosis in different T cell subsets including DNT cells. We hope to clarify this point in our next study. We have now added the following description in the current Discussion section to address the reviewer’s concern.

“Third, whether DNT cells would be deleted by the TNF expressed by myeloid cells is not understood clearly. Although we proposed the mechanisms, direct evidence of apoptosis in DNT cells is lacking. To ascertain our proposed mechanisms, further analyses would be required, e.g. determination of apoptosis in DNT cells of lymph nodes or in-vitro analysis to determine the apoptotic pathway in DNT cells.”

7. Result 5, Fig. 7b: The authors stated that whereas Tnf and Tnfr1 mRNA levels in the lymph nodes tended to be elevated in Sh3bp2KI/+Faslpr/lpr mice (line 445 - 446). The tendency of mRNA expression of both Tnf and Tnfr1 seem to be a little bit increased in the Sh3bp2KI/+Faslpr/lpr mice, however, it does not seem to be statistically different between Sh3bp2+/+Faslpr/lpr and Sh3bp2KI/+Faslpr/lpr mouse groups, indicating that Tnf and Tnfr1 mRNA levels between two groups are not significantly different and that the authors’ hypothesis that apoptosis process in Sh3bp2KI/+Faslpr/lpr mice is increased is not correct based on these results.

Response: As pointed out by the reviewer, there is only a tendency of increase in mRNA expression of both Tnf and Tnfr1 in the Sh3bp2KI/+Faslpr/lpr group. For RNA extraction, we had used whole lymph node tissues, which would imply that stromal cells and other immune cells, besides the myeloid cells, contaminated the RNA samples. Since the myeloid cells occupy a relatively small fraction in lymph nodes, the effect from non-myeloid cells might have masked the impact of myeloid cells alone.

              Along with the increased levels of cleaved caspase-3 in Sh3bp2KI/+Faslpr/lpr mice (Fig. 7a), we considered the apoptotic process to be upregulated in lymph nodes of Sh3bp2KI/+Faslpr/lpr mice. However, in order to directly address the reviewer’s concern, we would prefer to examine TNF mRNA or protein expression in myeloid cells in tissues and TNFR1 mRNA or protein expression in lymphocytes in future analyses.

8. Result 5, Fig. 8b: TNF is one of the most popular pro-inflammatory cytokines expressed and secreted from the macrophages under inflammatory condition. Based on the authors’ hypothesis, Sh3bp2KI/+Faslpr/lpr ameliorates autoimmunity and is anti-inflammatory, however, TNF production was increased in these mice compared to the Sh3bp2+/+Faslpr/lpr mice. Although the authors provided reference 48, the authors still need to explain the possibility of this observation which is contradictory to the fact well established in this field.

Response: As recommended by the reviewer, we have added the description in the Discussion section with reference to the following papers (Ref. #55-57).

“Contrary to the protective effect of TNF in the development of SLE, some reports show that TNF contributes to the progression of SLE. Administration of anti-TNF antibody is reported to attenuate lupus phenotypes in a murine lupus model with Fasl mutation [55] and in an experimental lupus model [56]. Since TNF has dualistic suppressive and promotive effects on lupus condition [57], timing and duration of TNF exposure or inhibition seem to be critical.”

55.       Korner, H.; Cretney, E.; Wilhelm, P.; Kelly, J.M.; Rollinghoff, M.; Sedgwick, J.D.; Smyth, M.J. Tumor necrosis factor sustains the generalized lymphoproliferative disorder (gld) phenotype. The Journal of experimental medicine 2000, 191, 89-96.

56.       Segal, R.; Dayan, M.; Zinger, H.; Mozes, E. Suppression of experimental systemic lupus erythematosus (SLE) in mice via TNF inhibition by an anti-TNFalpha monoclonal antibody and by pentoxiphylline. Lupus 2001, 10, 23-31, doi:10.1191/096120301675275538.

57.       Kollias, G.; Kontoyiannis, D. Role of TNF/TNFR in autoimmunity: specific TNF receptor blockade may be advantageous to anti-TNF treatments. Cytokine & growth factor reviews 2002, 13, 315-321.

9. Result 5, Fig. 8d: The reviewer strongly recommends the authors to examine the phagocytosis activity of BMMs in various time points (shorter than 24 h or longer than 48 h) since it has been reported in many studies that active phagocytosis of macrophages occurs within 1 h.

Response: In our phagocytosis experiments, we had measured the phagocytic activities at 4 h, 24 h, and 48 h. The values at 4 h were quite close to baseline levels, indicating that phagocytosis had not substantially progressed at that time point. We have now included the 4-h data in Fig. 8d of the revised manuscript. We consider this revised figure to show the time course of progression in phagocytosis.

              Appropriate time points might vary depending on experimental settings. The increased progression of phagocytosis seen in our study is in accordance with a previous study (Teresa Gracia-Aguilar, et al. J Immunol Res. 2016;2016:3845247. doi: 10.1155/2016/3845247) that had examined phagocytosis of apoptotic cells at 2, 4, 12, and 24 h, and had shown increased phagocytosis with time.

10. The labeling of some figures is broken (Fig. 3a – c). Please make a correction properly.

Response: We apologize for the errors in figures; we have now corrected the labeling in Fig. 3, as suggested by the reviewer.

11. There are some typos and grammatical errors. Please go over the entire manuscript carefully and correct all typo and grammatical errors.

Response: We are sorry for the errors in the manuscript; as recommended by the reviewer, we have carefully checked the manuscript again, and additional English editing has been done by native English speakers at Editage, an English editing service. We hope the manuscript is now satisfactorily edited. The certificate of English editing is attached herewith.

Round 2

Reviewer 3 Report

The revised version of the article is acceptable for publication.